# Effect of Printing Orientation on the Mechanical Properties of Low-Force Stereolithography-Manufactured Durable Resin

Antonio Martínez Raya [1,*], Josué Aranda-Ruiz [2], Gastón Sal-Anglada [3], Sebastián Martín Jaureguizahar [4] and Matías Braun [5]

1. Department of Organizational Engineering, Business Administration and Statistics, Technical University of Madrid—Universidad Politécnica de Madrid (UPM), 28040 Madrid, Spain
2. Department of Continuum Mechanics and Structural Analysis, Carlos III University of Madrid—Universidad Carlos III de Madrid (UC3M), 28911 Madrid, Spain; jaranda@ing.uc3m.es
3. Centre Internacional de Mètodes Numèrics en Enyinyeria (CIMNE), Technical University of Catalonia—Universitat Politècnica de Catalunya (UPC), 08034 Barcelona, Spain; gsal@cimne.upc.edu
4. Institute for Research in Materials Science and Technology, National University of Mar del Plata—Universidad Nacional de Mar del Plata (UNMDP), Mar del Plata 7600, Argentina; sebastianjaureguizahar@fi.mdp.edu.ar
5. Department of Fluid Mechanics and Aerospace Propulsion, Technical University of Madrid—Universidad Politécnica de Madrid (UPM), 28040 Madrid, Spain; matias.braun@upm.es
* Correspondence: antoniomartinez@upm.es; Tel.: +34-91-0676046

**Featured Application: The strength and rigidity of the durable resin make it ideal for a wide range of applications, including prototyping and product development, automotive and aerospace components, medical devices, consumer goods, custom tooling, and educational models. Its strength and precision are precious in producing functional prototypes, nonstructural automotive parts, custom medical devices, and durable electronic enclosures. This versatile material speeds up design processes and improves the durability and functionality of end products in various industries.**

**Abstract:** This study presents the results of fracture toughness tests conducted on specimens obtained by additive manufacturing techniques, specifically using low-force stereolithography. The samples were manufactured from a transparent 3D printing material for biocompatible applications, the so-called BioMed Durable Resin, which is a Formlabs-patented polymer material that simulates the strength and rigidity of polyethylene. The selected toughness tests in this context were performed following the ASTM D5045-99 guidelines. All tests were conducted under controlled laboratory conditions at 23 °C and 50% relative humidity, ensuring adherence to the standard and the replicability of the experimental results. To investigate the influence of printing plane orientation, specimens were produced at three printing orientation angles (0, 45, and 90 degrees). These angles were selected to provide a comprehensive evaluation of the anisotropy effects in the material. They cover both extreme orientations (0° and 90°) and include an intermediate value (45°), allowing us to assess variations in mechanical behavior across a representative range of printing orientations, consistent with prior research in the field. The experimental tests yielded data on the crack resistance and energy release rate for each angle of orientation. There are various implications of the findings, beyond materials engineering, for applications in biomedicine. Indeed, this same approach opens the door to new research methods for manufacturing certified biocompatible materials from such durable resins. Finally, complementary issues such as related medical applications have been slightly addressed for future work, since biomedicine innovation clusters can contribute to accelerating growth in this crucial field for productive sector activity and the local business environment.

**Keywords:** additive manufacturing techniques; low-force stereolithography; biocompatible applications; materials engineering

## 1. Introduction

Additive manufacturing (AM), more commonly known as 3D printing, is precipitating a profound transformation in a multitude of industries [1–8]. This technology enables the fabrication of mechanical components through the deposition of successive layers of material, thereby integrating design and manufacturing processes in ways that were previously unfeasible. This integration has resulted in notable improvements in efficiency, a reduction in material waste, and the capacity to create complex bespoke geometries, prompting designers, manufacturers, and logistics providers to adapt their procedures. Furthermore, AM has illustrated its capacity to address significant social challenges, such as the rapid production of medical equipment during the global response to the SARS-CoV-2 pandemic [4]. This has involved meeting substantial global demand and contributing to the alleviation of health crises around the world [5–8].

The development of various AM techniques, including fused deposition modeling (FDM), selective laser sintering (SLS), and stereolithography (SLA), has broadened the scope of materials utilized in 3D printing to include plastics, metals, and ceramics [9–13]. This expansion has prompted a significant body of research that examines the mechanical properties of printed components. Most of the research has focused on FDM because of the rapid production speed and low cost that it offers. However, FDM has inherent limitations, including a lack of surface precision and the presence of internal voids, which are a consequence of the physical restrictions imposed by the diameter of the extruder nozzle [14–17].

Stereolithography (SLA) is an additive manufacturing technique that employs a laser to construct structures by selectively curing a photopolymer resin layer by layer. This method offers superior resolution and reduces the occurrence of internal voids, resulting in enhanced mechanical properties and a reduction in anisotropy in the final components. The resolution of commercially available SLA printers is typically in the range of tens to hundreds of micrometers, determined by the diameter of the laser and the vertical displacement between layers [18–22]. This resolution is significantly higher than that of FDM, offering notable advantages in terms of precision and detail.

The effects of printing orientation on the mechanical properties of SLA-printed components have been the subject of recent research. The optimization of printing orientation has been shown to improve the tensile strength, compressive strength, and flexural modulus of low-force stereolithography (LFS) resin samples [23–25]. Furthermore, it has been demonstrated that reducing the layer thickness in SLA printing can enhance the strength of the resulting parts. Furthermore, studies have demonstrated that the incorporation of graphene fillers can significantly improve the stiffness and strength of stereolithographically printed materials when the printing orientation is optimized [26].

A comprehensive understanding of biocompatible materials' fracture toughness and energy release rate is essential for the reliable design of biomedical components. These properties assist in the prediction of how materials will perform under stress, which is of vital importance for ensuring their durability and safety in medical applications. By optimizing these properties, our study contributes to developing more effective and dependable biomedical devices, which ultimately enhance their performance and longevity in clinical settings.

Moreover, an understanding of the mechanical behavior of these materials is essential for the development of numerical tools that facilitate the optimization of such materials. A plethora of numerical models demonstrate considerable promise in elucidating the mechanical behavior of these materials [27].

Chen and Lu [28] studied the effect of build orientation on surface quality in rapid prototyping and found that as layer thickness decreases, traditional concerns about surface roughness due to inclined planes may diminish, and scan orientation within layers becomes a more important factor. This suggests that optimizing scan orientation may be critical to improving the surface quality of 3D printed parts. Wang et al. [10] have further demonstrated the significant impact of printing orientation on the mechanical properties of SLA-printed resin samples, emphasizing the necessity for meticulous optimization to

achieve optimal tensile strength, compressive strength, and flexural modulus. In particular, the influence of the rotation angle relative to the x, y, and z axes was subjected to a comprehensive investigation, revealing that strategic adjustments in these parameters can result in substantial improvements in mechanical performance. In a related study by Martín-Montal et al. [29], an extensive experimental methodology framework was proposed for the study and characterization of materials printed by stereolithography (SLA). The authors' work concentrated on tensile and compression tests at varying strain rates, intending to elucidate the impact of printing parameters, including printing angles (0–90 degrees) and layer heights (100 μm and 50 μm). Moreover, the effects of curing time and temperature on the mechanical behavior of SLA-printed materials were examined.

Current research on the mechanical properties of additively manufactured materials, particularly those produced via stereolithography, has focused primarily on quasi-static tensile and compressive tests. However, limited attention has been paid to the fracture toughness and energy release rate of these materials, especially in biocompatible applications. This knowledge gap is critical, as these properties are essential for ensuring the durability and safety of components in medical devices. Our study seeks to address this by evaluating the fracture toughness of LFS-printed biocompatible materials, contributing valuable insights to the field of biomedical engineering.

This study aims to examine the fracture behavior of specimens produced using low-force stereolithography. In particular, the study determines the fracture toughness of compact polymer samples produced using Durable Resin by FormLabs (Somerville, MA, USA) [30], a material that emulates the strength and rigidity of polyethylene and is certified as biocompatible. The study examines the impact of print orientation by analyzing samples printed at 0°, 45°, and 90°. Fracture toughness tests were conducted to determine the material's fracture toughness and energy release rate, in addition to the initial stiffness and peak load derived from the experimental data.

Notwithstanding the considerable progress made in understanding the mechanical characteristics of AM materials, several constraints persist. Most studies have focused on quasi-static tensile and flexural tests, with relatively little attention paid to the dynamic performance or the impact of varying manufacturing parameters on mechanical behavior. Although this study focuses on quasi-static tests to evaluate fracture toughness and energy release rate, dynamic tests such as impact and fatigue tests are planned for future research to provide a more comprehensive understanding of the material's behavior. It is recommended that future research be expanded to include a broader range of testing conditions and materials, as well as the development of constitutive models to predict the behavior of 3D printed materials better. Implementing such models would facilitate using numerical tools to design materials more efficiently, optimize mechanical performance, and expand the practical applications of AM technologies.

## 2. Experimental Setup

This study outlines a structured experimental approach to investigate and characterize materials produced using stereolithography. The aim is to address key considerations that can enhance the broader application of these materials. In SLA, a laser is directed onto a build platform, which selectively solidifies areas of a liquid photopolymer resin. As the laser traces the desired pattern, it initiates curing only in the regions exposed to its path, layer by layer, to form the final specimen. The curing process occurs in the plane of the platform and occurs layer by layer. Structures that facilitate the proper growth of the sample and its attachment to the building bed support the sample.

The resolution of the printing process is dependent on two factors: the diameter of the laser used and the increment of the vertical displacement between each layer, which the user can select. In stereolithography-based printers currently on the market, the aforementioned resolution ranges from hundreds of micrometers to dozens.

The material analyzed is Durable Resin from FormLabs [30]. The same company uses the Form 3 printer for fabrication. To produce the finished product, its geometry

must be in the STL (Standard Tessellation Language) file format. The printing software (PreForm 3.34.3) will use this file format to define specific parameters, such as the thickness of the layer (in this study, a thickness of 50 μm has been set), the orientation of printing, and the arrangement of the supports. These devices should ensure proper fixation of the samples during the process without jeopardizing their integrity, appearance, or final usability. This required a greater level of caution, since fatal errors in specimen preparation may lead to structural damage.

The printing process involves storing the base material in liquid form and heating it in a tank located in the printing area above the set of lenses and mirrors. The printer also uses a high-power ultraviolet (UV) laser to polymerize the UV-curing resin layer by layer selectively. Once the printing process is complete, the sample is submerged in an isopropyl alcohol solution with a concentration of 90 percent for a time determined by the manufacturer (15 min for durable specimens) to eliminate any residual uncured resin from the sample. Subsequently, the specimen can be post-cured using a UV heating device that provides both temperature and UV radiation. This process further solidifies the material, increasing its degree of polymerization and inducing cross-linking of the polymer chains. The resin manufacturer provides specifications regarding the requisite temperature and curing time. In this instance, the radiation source with a wavelength of 405 nm and a power output of 100 W is employed to maintain a constant temperature of 60 °C for 60 min, based on the resin manufacturer's recommendations for optimal curing of the Durable resin. This wavelength is widely used in stereolithography due to its efficiency in initiating polymerization in common photopolymers. Compared to other wavelengths (e.g., 365 nm or 450 nm), 405 nm offers an effective balance between energy absorption by the photoinitiators and curing efficiency, leading to improved mechanical properties in the printed parts.

The anisotropy characteristic of parts printed using fused deposition modeling is significantly reduced when employing the low-force stereolithography (LFS) technique. Minor differences in mechanical properties become apparent only once the elastic limit of the material is exceeded. Based on the characterization tests presented by Martín-Montal et al. [29], the material used in this study exhibits an elastic modulus of approximately E = 362.5 MPa, yield strength of $\sigma_y$ = 20 MPa, and a Poisson ratio of $\nu$ = 0.35. These values are consistent with those reported for similar SLA-printed resins, where elastic moduli typically range between 300 MPa and 400 MPa and yield strengths vary between 18 MPa and 25 MPa, as observed in other studies, such as those by Wang et al. [10]. The variations in these mechanical properties concerning printing angle were not taken into account, as they were less than 10 percent, in agreement with prior findings [29].

This work presents the calculation of the fracture stiffness (critical value of the Stress Intensity Factor, $K_{Ic}$) and the Critical Energy Release Rate $G_{Ic}$. At the onset of fracture, both in the mode I deformation and the opening mode, the calculation was carried out according to the ASTM D5045-99 standard [31], using the compact specimen geometry illustrated in Figure 1 and with the dimensions specified below, specifically W = 40 mm, a = 19 mm, d = 2 mm, and B = 10 mm.

The compact samples of the study according to three different printings were analyzed by defining the angle formed from the printing base with the initial notch plane. It has also been verified that the direction of printing did not influence the mechanical properties obtained. To gain insight into the effects of the angle on the material under study, three representative angles were selected: 0°, 45°, and 90° (see Figure 2). The 0° and 90° orientations represent extreme cases, where more significant differences in mechanical behavior are expected, if present. The 45° orientation serves as an intermediate case, which allows for an assessment of mechanical properties in a less extreme scenario. This combination of angles provides a comprehensive understanding of how different printing orientations affect the material's behavior. The test results are presented below in a proper summary.

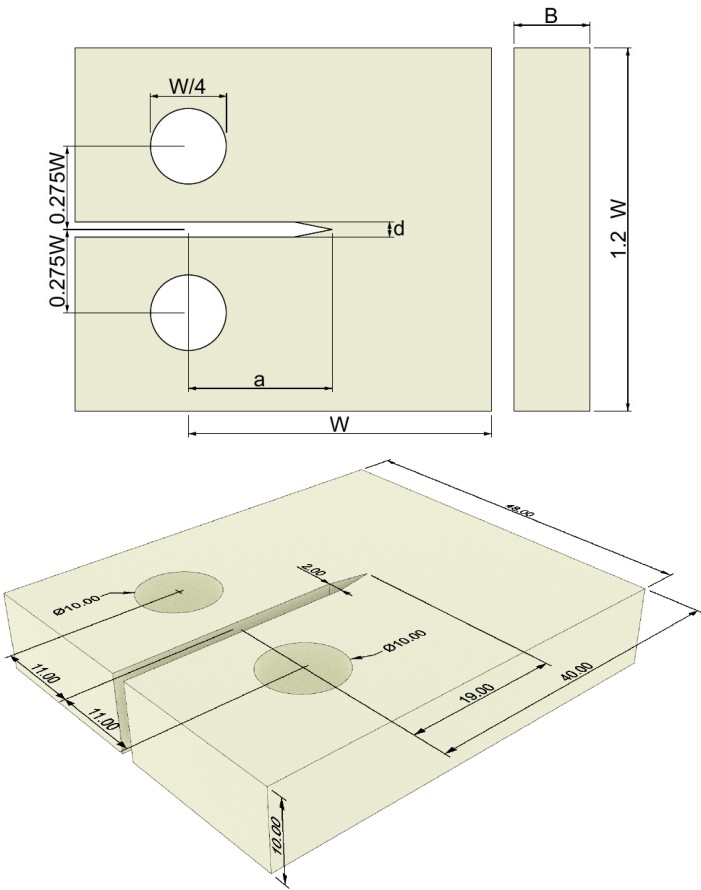

**Figure 1.** Compact specimen according to the ASTM D5045-99 standard [31]. Geometry parameters (**up**) and perspective view (**down**) with the dimensions used in the experiments. All sizes are shown in millimeters (mm). Source: own elaboration plotted with Inkscape (v. 1.3.2).

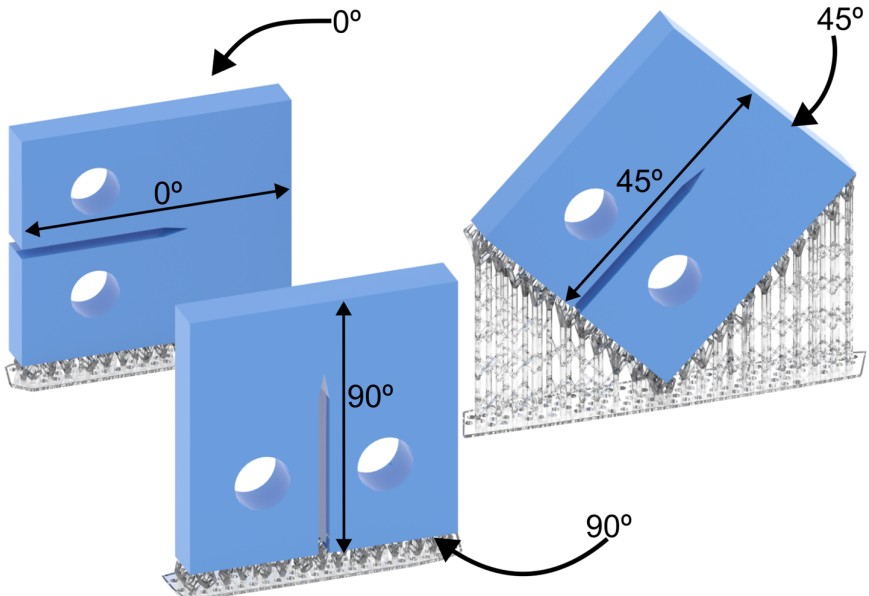

**Figure 2.** Three printing orientations for compact specimens. The printing orientation is drawn with a continuous line and dashed tangential. Source: own elaboration plotted with Inkscape (v. 1.3.2).

Once printed, the samples were machined to create a sharp initial notch. To machine the notch tip, a blade with a 0.5 mm radius was utilized, following the guidelines outlined

in the aforementioned standard. Figure 3 illustrates the blade employed to create the notch tip. In addition, compact tensile tests were performed on an Instron 8800 Universal Tester (Instron, Norwood, MA, USA) under crosshead displacement control at a rate of 5 mm/min. These tests were conducted at a temperature of 23 degrees Celsius. The three specimens were tested for each orientation. To ensure accuracy and minimize potential sources of error, the universal testing machine provided precise load and displacement measurements, and the samples were carefully aligned to avoid misalignment. Each test was repeated three times for each orientation to ensure consistency and reduce variability. Additionally, the compliance check method, as per ASTM D5045-99, was used to validate the accuracy of the results. In the next section, the results of the tests are presented in a test summary. The stresses in the selected specimens were also calculated, and they are summarized in the section below.

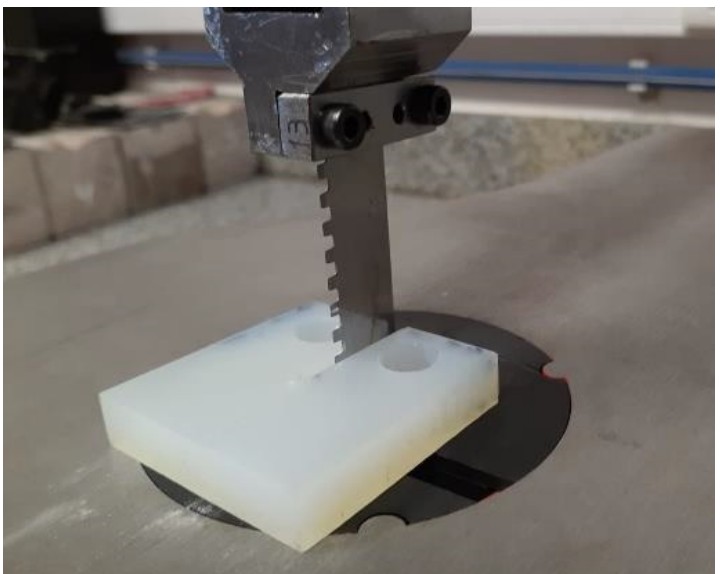

**Figure 3.** Blade with 0.5 mm Radius Used for Machining the Notch Tip According to ASTM D5045-99 Guidelines. Photo credit: authors.

## 3. Results and Discussion

Figure 4 illustrates the force–displacement curves of the compact specimens tested for each of the printing orientation angles under consideration.

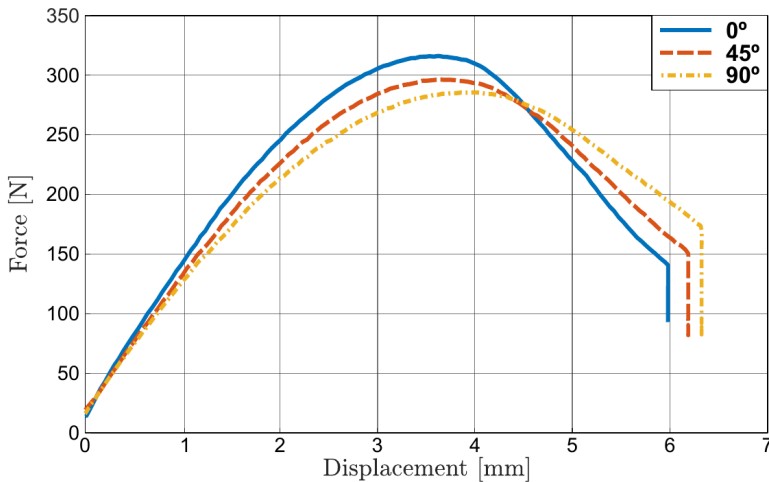

**Figure 4.** Load–displacement curves from compact tension tests conducted for each printing orientation. Source: own elaboration based on calculation results with Spyder (v. 5.5.1).

The results of this graphic allow for the calculation of several key parameters, including the initial stiffness and maximum load of the curves. The initial stiffness, calculated as the slope of the initial linear portion of the curve, indicates the resistance of the material to elastic deformation. The maximum load represents the maximum force sustained by the material before failure.

Table 1 presents a summary of the initial stiffness and maximum load of the force–displacement curves for different print orientations. The aforementioned values correspond to the mean values of the three samples tested for each case.

**Table 1.** Mechanical properties values derived from the force–displacement curves [1].

| Specimens | 0° | 45° | 90° |
|---|---|---|---|
| Stiffness [N/mm] | 129.68 | 117.90 | 111.67 |
| Peak load [N] | 316.01 | 296.13 | 285.50 |

[1] Depending on the orientation of each print on the selected specimens. Source: own elaboration based on calculation results with Spyder (v. 5.5.1).

The results indicate relatively consistent stiffness across the different orientations, suggesting a similar performance in terms of material resistance to deformation. The peak load values exhibit moderate variability, indicating that the material can sustain a relatively consistent maximum force. Notably, while there is a reduction in both stiffness and peak load as the pressure plane changes, this difference is approximately 10%, consistent with the results presented in [29].

The observed variability of approximately 10% in the mechanical properties can be attributed to certain factors associated with the manufacturing process of the specimen. In particular, the curing process may exert an influence on the ultimate characteristics of the printed specimens. The generation of the notch using the blade during the tests represents another factor that could potentially influence the results. This dispersion is consistent with the findings reported by other authors in the literature [29], indicating that such variability is not uncommon in similar studies.

Furthermore, it is crucial to emphasize that these tolerances are deemed acceptable, particularly in biomechanical applications, where minor discrepancies are unlikely to significantly impact the overall performance and functionality of the materials in question.

The value of the structural toughness of the material can be determined based on previous records, using the following expression [31].

$$K_Q = \frac{P_Q}{BW^{\frac{1}{2}}} \; f(x) \tag{1}$$

If the function $f(x)$ is calculated as follows, where $x = a/W$:

$$f(x) = 6x^{1/2} \frac{1.99 - x(1-x)\left(2.15 - 3.93x + 2.70x^2\right)}{(1+2x)(1-x)^{3/2}} \tag{2}$$

To calculate $P_Q$ according to the specified standard, first determine the initial compliance C by loading the specimen and drawing a best-fit straight line (AB) on the load–displacement curve. The compliance C is the reciprocal of the slope of this line. Then, draw a second line (AB8) with compliance 5% greater than that of line AB. If the maximum load *Pmax* sustained by the specimen falls between lines AB and AB8, use $P_{max}$ to calculate $K_Q$. If *Pmax* falls outside this range, determine $P_Q$ as the intersection of line AB8 with the load curve. It is important to ensure that the condition $P_{max}/P_Q < 1.1$ is met to validate the test. Table 2 presents the values of $P_Q$ for each of the samples tested, along with the mean values and standard deviation.

**Table 2.** $P_Q$ values for durable resin material (expressed in N) [1].

| Specimens | 0° | 45° | 90° |
|---|---|---|---|
| 1 | 298 | 278 | 317 |
| 2 | 264 | 256 | 267 |
| 3 | 288 | 308 | 273 |
| Mean | 283.3 | 280.7 | 285.7 |
| SD | 17.5 | 26.1 | 27.3 |

[1] Depending on the orientation of each print on the selected specimens. Source: own elaboration based on calculation results with Spyder (v. 5.5.1).

Finally, it is necessary to verify that the value of the expression $2.5\left(K_Q/\sigma_Y\right)^2$ is less than the thickness of specimen B, less than the initial notch length $a$, and less than the ligament value $(W - a)$. Notably, $\sigma_Y$ is the value of the material's yield strength. Thus, after evaluating this validity, $K_Q$ can finally be established as the value of the fracture toughness of the material: $K_Q = K_{Ic}$.

Table 3 shows the results of each test performed, including the average value of $K_{Ic}$ for each printing orientation considered. This allows the establishment of a final average value of $K_{Ic} = 1.32$ MPa·m$^{1/2}$ with a maximum standard deviation (SD) of 0.0687.

**Table 3.** Fracture toughness values for durable resin material (expressed in MPa·m$^{1/2}$) [1].

| Specimens | 0° | 45° | 90° |
|---|---|---|---|
| 1 | 1.34 | 1.25 | 1.32 |
| 2 | 1.27 | 1.34 | 1.29 |
| 3 | 1.39 | 1.29 | 1.32 |
| Mean | 1.33 | 1.32 | 1.31 |
| SD | 0.0579 | 0.0687 | 0.0182 |

[1] Depending on the orientation of each print on the selected specimens. Source: own elaboration based on calculation results with Spyder (v. 5.5.1).

The critical energy release rate $G_{Ic}$ values for each test were calculated using the following equation [31]:

$$G_{Ic} = \frac{\left(1 - v^2\right) K_{Ic}^2}{E} \tag{3}$$

Table 4 presents the values of the critical energy release rate at the time of fracture for each of the samples tested, along with the mean values and standard deviation.

**Table 4.** Critical energy release rate values for the durable resin material (expressed in $J/m^2$) [1].

| Specimens | 0° | 45° | 90° |
|---|---|---|---|
| 1 | 4.31 | 3.75 | 4.23 |
| 2 | 3.93 | 4.33 | 4.02 |
| 3 | 4.67 | 4.61 | 4.20 |
| Mean | 4.31 | 4.23 | 4.15 |
| SD | 0.3734 | 0.4357 | 0.1150 |

[1] Depending on the orientation of each print on the selected specimens. Source: own elaboration based on calculation results with Spyder (v. 5.5.1).

Figure 5 shows the fracture planes observed after testing the samples in different orientations: 0°, 45°, and 90°. The direction of propagation of the crack is from left to right. The fracture planes are straight, located on the symmetry plane, and follow the normal direction of the applied load. The fracture planes observed in the specimens oriented in the three analyzed directions appear relatively similar, suggesting that the material exhibits consistent fracture behavior. Upon closer examination, however, distinct regions can be

observed, exhibiting certain preferences concerning crack orientation. The aforementioned regions are indicated in each photograph by a circle, thereby signifying the orientation of the cracks.

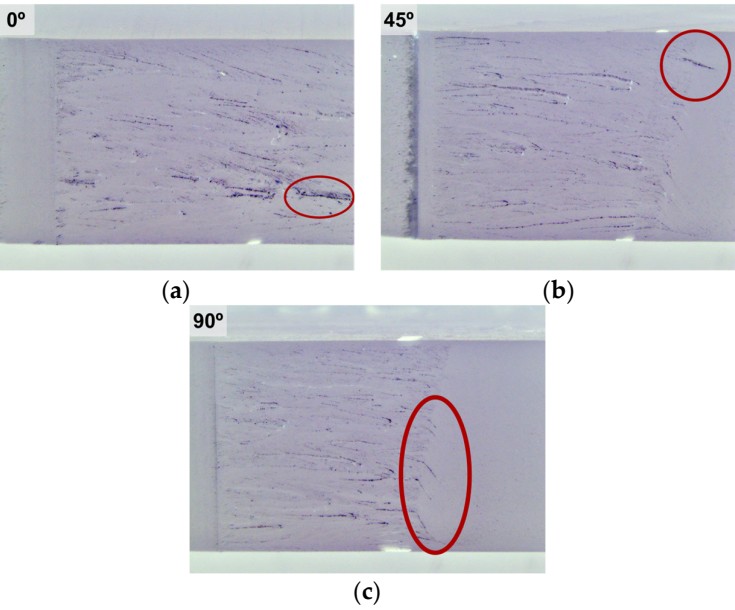

**Figure 5.** Testing the fracture patterns of specimens with (**a**) orientations at 0 degrees; (**b**) orientations at 45 degrees; and (**c**) orientations at 90 degrees. Photo credit (each): authors.

## 4. Conclusions

This study investigates the effect of print orientation angles on the fracture toughness and the critical energy release rate of durable resin specimens produced via low-force stereolithography. Fracture toughness tests were conducted on samples printed at 0, 45, and 90 degrees according to ASTM D5045-99 standards.

The results indicate that print orientation has a minimal impact on the fracture toughness and critical energy release rate of the resin. While slight variations in maximum load and stiffness were noted, these differences are not significant enough to affect the overall performance of the material.

This finding suggests that factors such as process efficiency and design geometry can be prioritized over print orientation without compromising mechanical integrity. This flexibility is advantageous for optimizing production and adapting designs to project-specific needs, potentially improving operational efficiency and reducing costs.

Moreover, this flexibility is highly beneficial across various industries, such as automotive, aerospace, healthcare, and consumer goods, where customized designs, minimal material waste, and mechanical reliability are crucial. The ability to produce complex geometries quickly while maintaining structural integrity accelerates prototyping, improves product durability, and reduces overall costs. Additionally, these results have implications beyond biomedical applications, as they may be relevant for other industries requiring durable materials, such as the automotive and aerospace sectors. The study also paves the way for further innovation in biocompatible materials, encouraging exploration of new manufacturing techniques and material combinations to enhance printed component properties.

In conclusion, this study makes a significant contribution to the understanding of print orientation effects and paves the way for future research into the effects of different materials and printing parameters on the performance and application of 3D-printed components in diverse fields.

## 5. Limitations of the Research and Future Directions

The study was confined to examining samples produced from BioMed Durable resin, a FormLabs polymer material that emulates the strength and rigidity of polyethylene, with layer thicknesses of 50 microns. Additionally, the tests were conducted with the sole objective of calculating the fracture toughness in Mode I. This focus on a specific material and fracture mode inevitably entails certain limitations in the broader understanding of the material's behavior under varied conditions.

As can be seen from Figure A1 in Appendix A, the research methodology proposed in the empirical study for examining and characterizing materials printed by stereolithography can be extended to other resins. This provides a scalable research solution for stakeholders interested in enabling progress in the understanding of additive manufacturing techniques based on low-force stereolithography.

To build on this research, further tests must be performed to estimate the fracture toughness for modes II and III at different orientation angles of the print. An expanded scope would facilitate a more comprehensive understanding of the material's performance under different loading conditions and orientations. Furthermore, an investigation of the fracture toughness of the material when subjected to alternative resins and changes in printing parameters, such as layer thickness, would facilitate a deeper understanding of the impact of these variables on mechanical properties.

To complement these efforts, dynamic tests such as impact and fatigue will be incorporated to provide a more comprehensive understanding of the material's performance in real-world applications. Additionally, further experiments will be conducted to explore the effects of varying layer thicknesses, printing speeds, and post-curing parameters on the mechanical properties of the resin. These follow-up studies will help refine the optimization of printing parameters for different industries and expand the practical applications of LFS-printed resins.

Furthermore, future research could investigate the long-term durability and biocompatibility of BioMed Durable Resin in authentic medical applications. Studies on the interaction of this material with biological tissues, its response to sterilization processes, and its performance under physiological conditions would provide critical data for its practical use in medical devices. Moreover, collaboration with medical professionals and industry experts can facilitate the identification of novel applications and design innovations, thus accelerating the adoption of 3D-printed biocompatible materials in healthcare.

In conclusion, these prospective avenues of inquiry will not only address the shortcomings of the present study but also pave the way for advances in both material science and biomedical engineering, thereby fostering innovation and improving the efficacy of 3D-printed medical solutions.

**Supplementary Materials:** The following supporting information can be downloaded at: https://www.mdpi.com/article/10.3390/app14209529/s1.

**Author Contributions:** Conceptualization, M.B., A.M.R. and J.A.-R.; methodology, M.B., J.A.-R. and G.S.-A.; software, J.A.-R.; validation, J.A.-R., G.S.-A. and S.M.J.; formal analysis, G.S.-A.; investigation, S.M.J.; resources, J.A.-R.; data curation, M.B.; writing—original draft preparation, M.B. and A.M.R.; writing—review and editing, A.M.R.; visualization, M.B. and A.M.R.; supervision, A.M.R.; project administration, A.M.R. All authors have read and agreed to the published version of the manuscript.

**Funding:** The authors are grateful for the support of the Regional Government of Madrid and the Ministry of Science, Innovation, and Universities of the Kingdom of Spain with the funding provided by the European Union (EU) to M.B. from NextGenerationEU (PRTR-C17.I1).

**Institutional Review Board Statement:** Not applicable.

**Informed Consent Statement:** Not applicable.

**Data Availability Statement:** Data is contained within the article and Supplementary Materials.

**Acknowledgments:** The authors wish to acknowledge the support in using materials for experiments at the UC3M facilities located in Leganés, Spain.

**Conflicts of Interest:** The authors declare no conflicts of interest.

## Appendix A

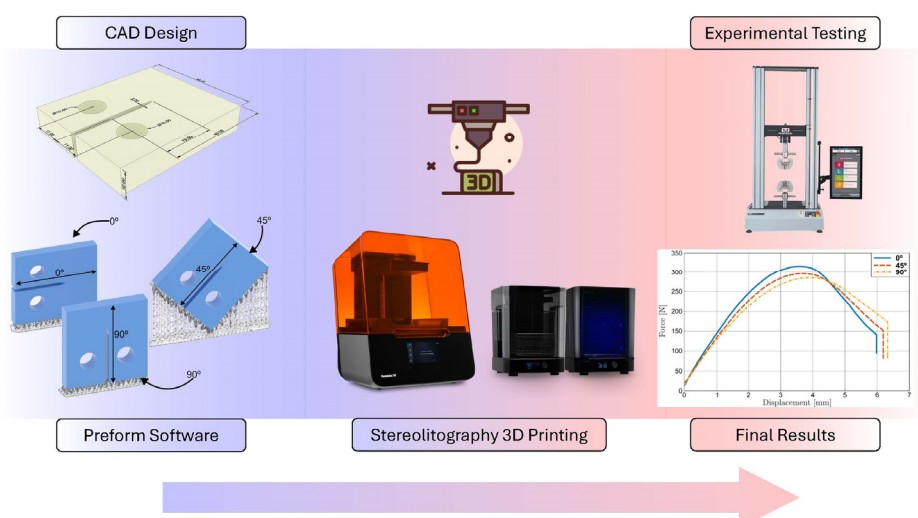

**Figure A1.** Graphical overview of the research methodology. Source: own elaboration.

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
