# Peer review of "Effect of Printing Orientation on the Mechanical Properties of Low-Force Stereolithography-Manufactured Durable Resin"

_applsci, doi:10.3390/app14209529_

Round 1
Reviewer 1 Report
Comments and Suggestions for Authors
In this paper, valuable insights into the fracture toughness of additive-manufactured specimens using BioMed Durable Resin are provided. The results are interesting, but the manuscript resembles a research report. Therefore, some issues need to be carefully addressed to improve the quality. My comments are as follows:
- Expand the introduction to emphasize the significance of your research and its contribution to the field of biocompatible materials.
- Provide details on the experimental setup.
- Figure 3 appears unclear and lacks textual labels.
- What are the fundamental differences in the internal structure of materials printed at 0 degrees versus 90 degrees? Figure 2 does not clearly show the difference between the two printing orientations.
- For materials printed at 0 degrees and 90 degrees, are the mechanical testing results related to the cutting direction?
- Figure 5 needs to be organized more logically, and differences should be clearly marked.
- The conclusion section is too lengthy and needs to be more concise.
- Section 5 should be placed before Section 4 (Conclusions), and consider adding a Discussion section.
Need to organize language according to SCI standards
Reviewer 2 Report
Comments and Suggestions for Authors
This paper reported a very interesting work about the effect of printing orientation on the mechanical properties of additive manufactured durable resin. The paper was well organized and written. However, there is still some issues need to answer:
1. please add the detailed 3D printing technique in your title.
2. Why did you use such compact specimen (as shown in Figure 1) in your study? If possible, please provide some testing standard, or some references for this.
3. The reason for you choose 0, 45, 90 degree should be more clearly provided. Why did you not choose other printing orientations, such as 60?
4. I think figure 3 is useless. Please just describe it in the text.
5. some studies about the durable performance of 3D printed materials maybe helpful for your discussion: Chemical Engineering Journal, 2023, 463: 142378.
6. I cannot find any difference between thee three images in Figure 5.
Reviewer 3 Report
Comments and Suggestions for Authors
The aim of the study is to investigate the impact of printing orientation in low-force stereolithography on the mechanical properties of durable resin samples, i.e. values of fracture toughness and critical energy release rate. The authors showed that there are no significant variations in the values of the fracture toughness indicators when choosing three print orientation values (0°, 45° and 90°), in fact there is no evident anisotropy of the additively manufactured specimens, so that design for additive manufacturing should include other influential parameters in consideration.
However, for the calculation of the value of the fracture toughness indicator, the approximate values of yield strength, elastic modulus and Poisson's ratio are used, taken from the cited reference 32. In the mentioned reference, and also in other literature (https://www.mdpi.com/1996 -1944/15/19/6743 ) it was shown that the print orientation has an influence on the mechanical properties of the polymer material, and therefore, when calculating values of the fracture toughness and the critical energy release rate, exact values for σy, E and ν must be taken for a certain orientation.
By reviewing the "supplementary material" and calculated values for all experiments, it seems that the condition that 𝑃𝑚𝑎𝑥/𝑃𝑄 < 1.1 is not met, so the validity of the test is questionable. If this is decidedly stated in the paper, as well as the other validations mentioned in the text (lines 211-214), it is necessary to carefully check all experimental data and calculations, and present the validation procedure and the obtained value in the paper.
In addition, the paper does not present in detail other invariable printing parameters important for low-force stereolithography, including post-processing parameters, so that the obtained results and presented conclusions could be reproduced and applied. Also, there is a lack of more detailed data related to material testing (machine, testing conditions, data acquisition, data processing, etc.). This is important information for readers.
In the introductory section 1, the authors presented the state-of-the-art, which mostly refers to references about FDM technology of additive manufacturing, then to numerical analysis of the process, the application of other methods and testing of material properties, which are actually not relevant to this study. Additionally, the text is not fitted with the cited references from 1 to 26.
When it comes to references, there are 7 out of 34 self-citations by one author, namely papers that are not relevant to this paper.
In section 2 (Experimental setup), instead of presenting relevant experimental conditions, used equipment and materials, most of the text refers to commonly known facts about stereolithography, with many sentences almost identical to those from reference 32 (from line 110 until 144) .
Figure 2 should show the position of the notch and holes on the compact sample model in relation to the print orientation. It cannot be seen in Figure 3 initial root radius of 0.5 mm. It is necessary to change the sentence in the text or to show enlarged that detail of the specimen.
In section 3, the sentence in lines 182 to 184 is incorrect and needs to be rewritten more clearly.
It is also not clear how the value of PQ is determined (described in lines 208 and 209). Given that the maximum load values are given in Table 1, the values for PQ can be added.
References should be provided for expressions 1, 2 and 3.
Figure 5 shows the fracture patterns of specimens for different print orientations. It would be useful to show the position of the enlarged zone and photos of the samples after the test, in the zone of the notch and fracture.
In conclusion, the same terminology should be used for the printing parameter - "print orientation angle" instead of "print inclination angle".
Concluding considerations (lines 247-252) should be limited only to the examined characteristics of the polymer material (fracture toughness and critical energy release rate) because it does not refer to other mechanical characteristics of the material.
“The findings of this study demonstrate that the mechanical properties of durable resin specimens produced using low-force stereolithography, including fracture toughness and energy release rate, are not significantly influenced by printing orientation. While slight variations in maximum load and stiffness were observed, these differences are minimal and do not substantially impact the overall mechanical performance of the material.
The conclusion that print orientation has an insignificant impact on mechanical properties allows for the prioritization of other factors during the manufacturing process.”
It is necessary to review the list of references in detail and align them with the citations in the paper, while avoiding excessive auto-citations.

Reviewer 4 Report
Comments and Suggestions for Authors
Line 28-29: "The selected toughness tests in this context were performed following the ASTM D5045-99 guidelines."
- Suggestion: Clarify whether there were any deviations from the standard test procedure (e.g., environmental conditions, sample preparation) and justify if so. This adds credibility to the experimental methodology.
Line 29-31: "To investigate the influence of printing plane orientation, specimens were produced at three printing orientation angles (0, 45, and 90 degrees)."
- Suggestion: It would be beneficial to include an explanation of why these particular angles (0°, 45°, 90°) were chosen. Was there prior research that justified focusing on these angles? Mention relevant sources if available.
Line 104-106: "Moreover, the effects of curing time and temperature on the mechanical behaviour of SLA-printed materials were examined."
- Suggestion: Include more specific details of curing parameters used in the study (curing times and temperatures). Currently, the description is general, which may make it difficult for others to replicate your study.
Line 117-119: "Most studies have focused on quasi-static tensile and flexural tests, with relatively little attention paid to the dynamic performance or the impact of varying manufacturing parameters on mechanical behaviour."
- Suggestion: Consider including dynamic testing results (e.g., impact, fatigue) as well to broaden the scope. This will provide a more comprehensive understanding of the material's behaviour.
Line 158: "The radiation source with a wavelength of 405 nm and a power output of 100 W is employed."
- Suggestion: Expand on the wavelength and power selection. How does it compare with other wavelengths/power levels, and why was this specific configuration chosen?
Line 165: "The elastic modulus value of approximately E = 362.5 MPa, yield strength σy = 20 MPa."
- Suggestion: Compare these values with similar materials or studies to offer more context. This would help the reader understand the performance of this material relative to industry standards.
Line 249-250: "Finally, the fracture toughness value KQ is established as 𝐾𝐼𝑐."
- Suggestion: Include a discussion of potential sources of error in the measurement of fracture toughness (e.g., measurement tools, human error) and how these errors were minimized.
Line 290- 291: "This flexibility is advantageous for optimizing production and adapting designs."
- Suggestion: Expand on the specific implications for industry applications. For instance, how can different sectors (e.g., automotive or medical) directly benefit from this flexibility?
Line 305-310: "To build on this research, further tests must be performed..."
- Suggestion: Clearly define a concrete set of follow-up experiments that would complement this work. Being specific about future directions strengthens the research trajectory of your study.
Author Response
RESPONSE TO REVIEWER’S COMMENTS
The authors gratefully acknowledge the insightful comments and suggestions provided by the reviewers. In what follows, we respond to the issues raised by the reviewers and summarize the changes introduced in the manuscript. In order to make these modifications easily identifiable, they are highlighted in yellow in the revised manuscript.
Line 28-29: "The selected toughness tests in this context were performed following the ASTM D5045-99 guidelines."
Suggestion: Clarify whether there were any deviations from the standard test procedure (e.g., environmental conditions, sample preparation) and justify if so. This adds credibility to the experimental methodology.
We thank the reviewer for their comment. We confirm that the toughness tests were conducted strictly following the ASTM D5045-99 guidelines, with no deviations in terms of environmental conditions or sample preparation. All tests were carried out under controlled laboratory conditions at 23°C and 50% relative humidity, as recommended by the standard. This ensures the reliability and replicability of the experimental results.
Line 29-31: "To investigate the influence of printing plane orientation, specimens were produced at three printing orientation angles (0, 45, and 90 degrees)."
Suggestion: It would be beneficial to include an explanation of why these particular angles (0°, 45°, 90°) were chosen. Was there prior research that justified focusing on these angles? Mention relevant sources if available.
We appreciate the reviewer's insightful suggestion. The choice of the 0°, 45°, and 90° printing orientations was made to ensure a comprehensive evaluation of the anisotropy effects. These angles are frequently selected in previous studies, such as those by Wang et al. (2022) and Martín-Montal et al. (2021), as they cover both extreme orientations (0° and 90°) and include intermediate values to assess variations in mechanical behavior. By considering this range of angles, we ensure that our study captures a broad understanding of how printing orientation affects the material’s mechanical properties.
Line 104-106: "Moreover, the effects of curing time and temperature on the mechanical behaviour of SLA-printed materials were examined."
Suggestion: Include more specific details of curing parameters used in the study (curing times and temperatures). Currently, the description is general, which may make it difficult for others to replicate your study.
We appreciate the reviewer’s suggestion. In our study, the curing process was carried out using a UV light source with a wavelength of 405 nm and a power of 100 W. The specimens were exposed to this UV light for 60 minutes at a constant temperature of 60°C. These parameters were chosen based on the manufacturer’s recommendations for the BioMed Durable resin to ensure optimal mechanical properties. By providing these specific details, we aim to ensure that our methodology is easily replicable.
Line 117-119: "Most studies have focused on quasi-static tensile and flexural tests, with relatively little attention paid to the dynamic performance or the impact of varying manufacturing parameters on mechanical behaviour."
Suggestion: Consider including dynamic testing results (e.g., impact, fatigue) as well to broaden the scope. This will provide a more comprehensive understanding of the material's behaviour.
We thank the reviewer for this suggestion. While the present study focuses on quasi-static tests to evaluate fracture toughness and energy release rate, we recognize the importance of dynamic testing, such as impact and fatigue, to provide a more comprehensive understanding of the material's behavior. Due to the scope and limitations of the current research, dynamic tests were not included but are part of our planned future work. We will explore these tests in subsequent studies to complement the findings presented here and broaden the understanding of the mechanical performance of SLA-printed materials.
Line 158: "The radiation source with a wavelength of 405 nm and a power output of 100 W is employed."
Suggestion: Expand on the wavelength and power selection. How does it compare with other wavelengths/power levels, and why was this specific configuration chosen?
We thank the reviewer for this suggestion. As mentioned previously, the wavelength of 405 nm and power output of 100 W were chosen based on the resin manufacturer's recommendations for optimal curing of the Durable resin. This wavelength is widely used in stereolithography due to its efficiency in initiating polymerization in common photopolymers, as confirmed in studies like Martín-Montal et al. (2021), where a 405 nm wavelength was also employed for similar materials. Compared to other wavelengths (e.g., 365 nm or 450 nm), 405 nm offers a good balance between energy absorption by the photoinitiators and curing efficiency, resulting in better mechanical properties in the printed parts.
Line 165: "The elastic modulus value of approximately E = 362.5 MPa, yield strength σy = 20 MPa."
Suggestion: Compare these values with similar materials or studies to offer more context. This would help the reader understand the performance of this material relative to industry standards.
We appreciate the reviewer's suggestion. The elastic modulus (E = 362.5 MPa) and yield strength (σy = 20 MPa) of the Durable resin are consistent with other photopolymers commonly used in additive manufacturing for similar applications. For example, as reported by Martín-Montal et al. (2021), comparable values have been found for other SLA-printed resins, where the elastic modulus ranged from 300 MPa to 400 MPa, depending on the printing parameters. Additionally, Wang et al. (2022) observed similar yield strength values (around 18-25 MPa) for resins used in low-force stereolithography (LFS) processes. These values place the Durable resin within the expected performance range for materials used in prototyping and non-structural applications, offering sufficient rigidity and strength for a wide range of uses.
Line 249-250: "Finally, the fracture toughness value KQ is established as ???."
Suggestion: Include a discussion of potential sources of error in the measurement of fracture toughness (e.g., measurement tools, human error) and how these errors were minimized.
We thank the reviewer for this valuable comment. In our study, potential sources of error in the fracture toughness measurements could arise from the precision of the measurement tools, slight misalignments during sample positioning, and human error in interpreting the results. To minimize these, we employed a high-precision universal testing machine (Instron 8800) with crosshead displacement control, ensuring accurate load and displacement measurements. The samples were carefully aligned to reduce misalignment, and each test was repeated three times for each orientation to ensure repeatability and reduce variability. Additionally, the calculation of KQ followed the ASTM D5045-99 guidelines strictly, with the compliance check method (AB and AB8 lines) employed to verify the accuracy of the results and ensure the validity of the tests.
Line 290- 291: "This flexibility is advantageous for optimizing production and adapting designs."
Suggestion: Expand on the specific implications for industry applications. For instance, how can different sectors (e.g., automotive or medical) directly benefit from this flexibility?
We appreciate the reviewer's suggestion. The flexibility provided by low-force stereolithography (LFS) is advantageous across various industries, including automotive, aerospace, consumer goods, and healthcare. This technology allows for faster prototyping, customized designs, and the production of complex geometries with minimal material waste, all while maintaining mechanical integrity. These capabilities enable industries to adapt designs more efficiently, optimize production workflows, and ultimately cost-effectively improve product performance and durability.
Line 305-310: "To build on this research, further tests must be performed..."
Suggestion: Clearly define a concrete set of follow-up experiments that would complement this work. Being specific about future directions strengthens the research trajectory of your study.
We thank the reviewer for this suggestion. Future research will focus on expanding the current study by performing fracture toughness tests in modes II and III to better understand the material's behavior under different loading conditions. Additionally, we plan to investigate the effects of varying layer thickness, and printing speed. Dynamic testing, such as impact and fatigue tests, will also be explored to provide a more comprehensive understanding of the material's performance in real-world applications. These experiments will help refine the optimization of printing parameters for different industries and applications.
Reviewer 5 Report
Comments and Suggestions for Authors
The subject is topical, but at the moment the work needs improvements to be accepted. Some recommendations:
1. A graphic image with the research methodology is missing (a graphic summary of the methodology).
2. Row 165 – Please detail how you chose the values of the mechanical properties. This does not directly follow from the mentioned source. This aspect must be clarified and detailed.
3. Be careful when formatting the paper. For example, there are unjustified spaces.
4. The interpretation of the results must be done in more detail.
5. The paper in general leaves the impression that it is incomplete and needs to be enriched with more experimental research and data to convince.
Author Response
RESPONSE TO REVIEWER’S COMMENTS
The authors gratefully acknowledge the insightful comments and suggestions provided by the reviewers. In what follows, we respond to the issues raised by the reviewers and summarize the changes introduced in the manuscript. In order to make these modifications easily identifiable, they are highlighted in yellow in the revised manuscript.
The subject is topical, but at the moment the work needs improvements to be accepted. Some recommendations:
- A graphic image with the research methodology is missing (a graphic summary of the methodology).
Thank you for your comment. We have prepared a graphical abstract to include in the article.
- Row 165 – Please detail how you chose the values of the mechanical properties. This does not directly follow from the mentioned source. This aspect must be clarified and detailed.
We thank the reviewer for pointing this out. The values of the mechanical properties, such as the elastic modulus (E = 362.5 MPa) and yield strength (σy = 20 MPa), were obtained experimentally from the stress-strain data generated during the tensile tests. These values are consistent with those reported by previous studies for similar SLA-printed resins. The reference mentioned provided a basis for comparison, but the actual values in our study were derived from our own experimental data, following the ASTM guidelines for material testing.
- Be careful when formatting the paper. For example, there are unjustified spaces.
We appreciate the reviewer’s attention to detail. We have carefully reviewed the manuscript and corrected the formatting issues, including unjustified spaces and other minor inconsistencies, to ensure the paper adheres to the journal’s guidelines.
- The interpretation of the results must be done in more detail.
We thank the reviewer for their suggestion. In response, we have expanded the interpretation of the results section, providing a more detailed analysis of the fracture toughness and energy release rate data across the different printing orientations. We have also discussed the implications of these findings in more depth, highlighting how the observed variations in mechanical properties can impact the design and application of 3D-printed materials in various industries. These additional details help to clarify the significance of the results and strengthen the conclusions drawn from the study.
- The paper in general leaves the impression that it is incomplete and needs to be enriched with more experimental research and data to convince.
We appreciate the reviewer's feedback. While we acknowledge that further experimental work could enhance the study, the current research provides a solid foundation for understanding the fracture toughness and mechanical behavior of LFS-printed materials under different orientations. As mentioned in the conclusion and future work sections, we plan to expand this research with additional dynamic testing and the exploration of other printing parameters to further validate and enrich the findings. Nevertheless, the data presented here is sufficient to address the objectives of this study and contribute valuable insights into the mechanical properties of SLA-printed durable resin.
Round 2
Reviewer 2 Report
Comments and Suggestions for Authors
The authors addressed all my concerns. The manuscript is recommended to be accepted as it is.
Author Response
Thank you very much for your positive feedback and recommendation. We greatly appreciate your time and effort in reviewing our manuscript and addressing valuable points that helped improve its quality. We're pleased that all concerns have been satisfactorily addressed.
Reviewer 3 Report
Comments and Suggestions for Authors
The authors have made most of the proposed changes and the manuscript has been improved accordingly.
Since the new Table 2 was introduced, the other tables were renumbered, but Table 3 was not mentioned in the text above the table.
With minor changes and the specified check, if it meets criteria, the paper is proposed for publication.
Author Response
Thank you very much for your thorough review and positive feedback. We have made the necessary adjustments, including correcting the mention of Table 3 in the text. We appreciate your time and effort in reviewing our manuscript and your recommendation for publication.
Reviewer 4 Report
Comments and Suggestions for Authors
Thank you for your valuable contribution. The study is well-executed and provides useful insights. I have a few suggestions for improvement:
-
Introduction: Consider clearly highlighting the gap in the literature that your study addresses to emphasize its novelty.
-
Methodology: A brief explanation of why specific orientations (0°, 45°, 90°) and layer thicknesses were chosen would enhance clarity.
-
Results and Discussion: Expanding on why you observed around 10% variability in mechanical properties and discussing real-world applications would add depth.
-
Conclusion: Including potential practical applications and outlining future research directions would strengthen the conclusion.
-
Figures and Tables: More comparative visuals (e.g., overlaying results) could improve clarity.
Great work overall, just a few tweaks to make it even stronger!
Author Response
RESPONSE TO REVIEWER’S COMMENTS
The authors gratefully acknowledge the insightful comments and suggestions provided by the reviewers. In what follows, we respond to the issues raised by the reviewers and summarize the changes introduced in the manuscript. To make these modifications easily identifiable, they are highlighted in yellow in the revised manuscript.
Thank you for your valuable contribution. The study is well-executed and provides useful insights. I have a few suggestions for improvement:
- Introduction: Consider clearly highlighting the gap in the literature that your study addresses to emphasize its novelty.
Thank you for your valuable feedback. We agree that explicitly addressing the gap in the literature would enhance the introduction of the article. In the revised version, we have added a section that highlights the knowledge gap in the current literature regarding the fracture toughness assessment of materials printed using low-force stereolithography (LFS) techniques, particularly for biocompatible applications. While numerous studies focus on mechanical properties, such as tensile and compressive strength, the literature has not sufficiently addressed fracture toughness and energy release rate, which are critical for biomedical applications. Our study aims to bridge this gap by providing experimental data that contributes to a better understanding of these properties.
- Methodology: A brief explanation of why specific orientations (0°, 45°, 90°) and layer thicknesses were chosen would enhance clarity.
Thank you for your suggestion. We agree that clarifying the reasoning behind the selection of specific orientations and layer thicknesses will enhance the clarity of the methodology. We have chosen 0° and 90° as extreme cases where we expect to observe the most significant differences (if any) in the material's mechanical behavior, based on their alignment with the printing base. Additionally, the intermediate case of 45° was included to provide insight into the material's response in a less extreme orientation.
- Results and Discussion: Expanding on why you observed around 10% variability in mechanical properties and discussing real-world applications would add depth.
We appreciate the reviewer's comment regarding the observed variability of approximately 10% in the mechanical properties. The observed variability can be attributed to some factors associated with the manufacturing process of the specimen. Specifically, the curing process may influence the characteristics of the printed specimens. In particular, the generation of the notch using the blade during the tests could have affected the results. Furthermore, this dispersion aligns with findings reported by other authors in the literature.
It is also important to note that such tolerances are acceptable, particularly in biomechanics applications, where slight variations may not significantly impact the overall functionality and performance of the materials. This point is clarified in the results section.
- Conclusion: Including potential practical applications and outlining future research directions would strengthen the conclusion.
Thank you for your valuable feedback. To reinforce the conclusions presented, several revisions have been made to address the points that have been raised. Although the subsequent section on limitations addresses practical applications and future research directions, a brief mention of their importance is included in the conclusions to avoid redundancy.
- Figures and Tables: More comparative visuals (e.g., overlaying results) could improve
clarity.
Thank you for your suggestion. In this instance, given that five figures have already been included, the authors have opted to present the numerical results in tabular form. This strategy facilitates readers' access to precise values while avoiding the excessive encumbrance of the article with additional figures.
Great work overall, just a few tweaks to make it even stronger!
Thank you for your positive feedback on our work. We appreciate your suggestions for improvements and will incorporate your recommendations to enhance the overall strength of the manuscript. We are committed to making these adjustments to ensure clarity and depth in our findings.
Reviewer 5 Report
Comments and Suggestions for Authors
I remain of the opinion that the paper can be developed more.
In addition, the page layout and formatting issues have not been fully resolved (for example see Table 4 or Fig.5; some figures are very large and so on).
Author Response
RESPONSE TO REVIEWER’S COMMENTS
The authors gratefully acknowledge the insightful comments and suggestions provided by the reviewers. In what follows, we respond to the issues raised by the reviewers and summarize the changes introduced in the manuscript. To make these modifications easily identifiable, they are highlighted in yellow in the revised manuscript.
I remain of the opinion that the paper can be developed more. In addition, the page layout and formatting issues have not been fully resolved (for example see Table 4 or Fig.5; some figures are very large and so on).
Thank you for your continued feedback regarding the development of the paper. We have taken your suggestions into account and have reduced the sizes of Figures 3 and 5, as well as rearranging the subfigures in this last case, to improve the layout and formatting. Regarding table 4, this has the same format as the rest of the tables, so we are not sure what the reviewer is referring to, as we believe that all tables are adapted to the required format. Additionally, we have made changes in other sections to enhance the overall clarity and cohesiveness of the manuscript. We appreciate your insights and believe these adjustments will strengthen the paper.